# Nanostructure Characteristics of Al_3_Sc_1−x_Zr_x_ Nanoparticles and Their Effects on Mechanical Property and SCC Behavior of Al–Zn–Mg Alloys

**DOI:** 10.3390/ma13081909

**Published:** 2020-04-18

**Authors:** Ying Deng, Ziang Yang, Guo Zhang

**Affiliations:** 1School of Materials Science and Engineering, Central South University, Changsha 410083, China; yangza39@csu.edu.cn (Z.Y.); 163112087@mail.csu.edu.cn (G.Z.); 2State Key Laboratory of Powder Metallurgy, Central South University, Changsha 410083, China

**Keywords:** aluminum alloys, nanostructure, microstructure, strength, stress corrosion cracking, nanoparticles

## Abstract

The Nanostructure characteristics of Al_3_Sc_1−x_Zr_x_ nanoparticles and their effects on the mechanical properties and stress corrosion cracking (SCC) behavior of Al–Zn–Mg alloys were investigated by 3D atom probe analyses, high-angle annular-dark-field scanning transmission electron microscopy methods, electron back scattered diffraction techniques, electrochemical measurements, slow strain rate tests and quantitative calculations. The results show that adding small amounts of scandium (0.10 percent by weight) and zirconium into Al–Zn–Mg extrusion bars can precipitate Al_3_Sc_1−x_Zr_x_ nanoparticles with a number density of (7.80 ± 3.83) × 10^21^ per cubic meter. Those particles, with a low lattice misfit with matrix (1.14 ± 0.03 percent) and stable core-shell L12-nanostructure in aged Al–Zn–Mg alloys, can increase the yield strength by 161 ± 7 MPa via strong Orowan strengthening (the theoretical calculated value is 159 MPa) and weak Hall-Petch strengthening (the theoretical calculated value is 6 MPa). Moreover, Al_3_Sc_1−x_Zr_x_ nanoparticles can change the fracture mechanism of alloys in 3.5% NaCl solution from intergranular cracks to transgranular failure, and decrease the proportion of high-angle grain boundaries from 87% to 31%, thus reducing the microchemistry differences around the grain boundaries and anodic dissolution kinetics, and improving intergranular SCC resistance and ductility. This study offers a new approach to the simultaneous improvement in mechanical property and corrosion performance of high strength alloys.

## 1. Instruction

Al–Zn–Mg alloys, due to their high-strength-weight, are applied widely in aerospace structural components and for highly stressed parts in automobiles and high-speed trains for fuel saving [1,2,3]. When airplanes or trains are in service in humid environments such as those present in marine, industry or rural atmospheres, 7xxx series aluminum alloys used as air-frame structures and car body members are exposed to mechanical loading, and thus, suffer from stress corrosion cracking (SCC) [2,4,5,6]. Adding high contents of alloying elements, such as Zn and Mg, is the main method to strengthen heat-treatable, ageing-hardenable aluminum alloys [7,8]. However, this accelerates the concentration of alloying elements and microchemistry heterogeneity on grain boundaries, further decreasing SCC resistance and ductility [9,10,11,12]. Therefore, it is challenging to achieve high strength, good ductility and satisfactory SCC resistance of ageing-hardenable 7xxx series aluminum alloys. 

Currently, the measures taken to improve the SCC resistance of 7xxx series aluminum alloys are mainly based on optimizing quenching and aging heat treatment processes [2,5], or adding rare earth elements to modify the grain boundaries [13,14]. T7 over-aging treatment [2,4,6], retrogression and re-aging (RRA) [15], dual-retrogression and re-aging (DRRA) [16] and step-quench and aging heat treatment (SQA) [6,16] have been reported to be effective in enhancing the SCC resistance of Al–Zn–Mg(–Cu) alloys. However, overaging to a T7 state would lead a loss of about 10–15% in strength, and due to the demand of a short retrogression or step-quench time, RRA, DRRA and SQA are not suitable for the large-scale fabrication using 7xxx series aluminum alloys with high quenching sensitivities. Regarding the effects of microalloying elements on the mechanical properties and SCC of 7xxx series aluminum alloys, there have been many relevant reports. On the one hand, Li [17] and our previous paper [18,19] illustrated that the addition of Sc could increase the strength of Al–Zn–Mg alloy plates by forming primary and secondary Al_3_Sc_1−x_Zr_x_ particles. On the other hand, the SCC performance of 7xxx series aluminum alloy plates can be improved by microalloying techniques. For instance, Bobby Kannan [6] and our previous work [20] showed that the SCC resistance of Al–Zn–Mg–Cu–Zr or Al–Zn–Mg alloys can be improved by adding 0.25 wt.% scandium, even in the peak aged state. Wang [21] indicated that Al–Zn–Mg–Cu–Zr alloys with Cr and Pr exhibited higher resistance to SCC than those without Pr- or Zr-containing dispersoids. However, to the best of the authors’ knowledge, there have been few attempts to enhance both the strength and SCC corrosion resistance of high-strength aluminum alloys by adding Sc. 

A high content of Sc (above 0.20 wt.%) in Al alloys can facilitate the formation of microscaled primary Al_3_Sc or Al_3_Sc_1−x_Zr_x_ particles, resulting in stress concentrations and deteriorating ductility [22,23,24]. Our previous work [18] further indicated that with the increase of Sc content, the elongation of Al–Zn–Mg alloys plates decreased. From this perspective, this paper aims to precipitate Al_3_Sc_1−x_Zr_x_ nanoparticles by adding small amounts of Sc (0.10 percent by weight) into Al–Zn–Mg alloys to improve ductility, strength and SCC resistance, thereby avoiding the negative effects of microscaled primary Al_3_Sc_1−x_Zr_x_ particles on ductility. Based on the improved performance, the nanostructure characteristics of Al_3_Sc_1−x_Zr_x_ nanoparticles in the T6 Al–Zn–Mg extruded alloy bars will be investigated. Unlike previous reports of the characterization of Al_3_Sc_1−x_Zr_x_ dispersoids, which either focused on composition distribution alone [25,26,27], or on their structure [28], or morphology [29], in this paper, all the characteristics of Al_3_Sc_1−x_Zr_x_ nanoparticles, including atomic distribution, lattice arrangement, the degree of coherence with matrix, number density and size, will be taken into consideration. Furthermore, the relationship between the aforementioned nanostructure characteristics of the particle and mechanical properties/SCC behavior will be established. This study offers a new method for the simultaneous improvement in the mechanical properties and SCC resistance of high-strength aluminum alloys.

## 2. Materials and Methods

A base Al–Zn–Mg ingot containing 5.20 Zn, 1.82 Mg, 0.29 Cu, 0.31 Mn, 0.09 Si and 0.18 Fe (wt.%), was fabricated by a semicontinuous casting process. The low additions of Cu reduce the risk of hot cracking during solidification and casting [18]. Some base alloys were added with 0.10 Sc and 0.10 Zr (wt.%). The ingots, with a diameter of 120 mm, were all subjected to step homogenization treatment (300 °C/12 h + step 470 °C/12 h) and air cooling. Then, the homogenized alloys were interannealed at 420 °C for 4 h and hot extruded to bars with a diameter of 13 mm. The total extrusion ratio was 22 and the extrusion temperatures ranged from 380–410 °C. Finally, the extruded alloys were subjected to solution-quenching-aging treatment (470 °C/1 h + water quenching + 120 °C/12 h). The aged alloys were investigated by mechanical, electrochemical and SCC tests, and microstructure observations. 

Slow strain rate tests (SSRT) were applied to evaluate SCC susceptibility according to GB 15970.7-2000 [30,31] and ASTM G129-00 [32] and performed on WDML-1 slow tension stress corrosion testing machine. The gauge length and diameter of the rod tensile specimens for SSRT were 40 mm and 6 mm, respectively. For comparative study, SSRT parallel samples were exposed to air and neutral 3.5 wt.% NaCl solution at an applied strain rate of 1 × 10^−6^ s^−1^. The tensile direction was perpendicular to the extrusion direction. A ZEISS EVO10-3412 field emission gun scanning electron microscope (SEM), operating at 20 kV, equipped with electron back scattered diffraction (EBSD) system, was used to investigate the fracture surfaces and the grain boundary characteristics. To characterize the ageing precipitates, grain boundaries and nanoparticles, a Titan G^2^ 60–300 high-resolution aberration-corrected transmission electron microscope (HRTEM), operating at 300 kV, with a high-angle annular-dark-field (HAADF) detector, was applied for HADDF scanning transmission electron microscopy (STEM) imaging. A LEAP 4000X HR under UV laser pulsing was used for three-dimensional atom probe tomography (3DAPT) of T6 samples, at a laser energy of 40 pJ and a pulse frequency of 250 kHz. The IM6ex electrochemistry workstation, with a three-electrode system (reference electrode: saturated calomel electrode (SCE), counter-electrode: platinum sheet, working electrode: about 1 cm^2^ aged samples) was used for electrochemical measurements, with a scan rate of 1 mV/s. 

## 3. Results 

### 3.1. Nanostructure and Atom Distribution of Al_3_Sc_1−x_Zr_x_ Particles

Figure 1 shows the tomography, composition and crystal structure of Al_3_Sc_1−x_Zr_x_ nanoparticles in the aged alloys. The Al_3_Sc_1−x_Zr_x_ particles are spherical and in nanoscale (Figure 1a). The atomic structures of the interface between Al_3_Sc_1−x_Zr_x_ and the aluminum matrix were further magnified and characterized by high-resolution HAADF STEM (Figure 1b–d). We observed that two sets of atomic planes, namely {100} and {010}, perfectly pass through the particles, indicating that Al_3_Sc_1−x_Zr_x_ nanoparticles have full lattice coherency with the matrix (Figure 1b). The fast Fourier transform (FFT) patterns, taken from the <001> zone axis, revealed that the aluminum matrix and Al_3_Sc_1−x_Zr_x_ both had an ordered of L1_2_ structure (Figure 1c–d). Moreover, the Al_3_Sc_1−x_Zr_x_ nanoparticles had twice the crystalline interplanar spacing of α(Al), and the lattice parameter misfit was calculated to be 1.14 ± 0.03 percent. L1_2_-Al_3_Sc_1−x_Zr_x_ particles had a cubic Cu_3_Al atom structure, whose space group was Pm-3 m. In a unit cell, one Al atom was surrounded by four Sc/Zr atoms (Figure 1e). The crystal structure was consistent with the high-resolution HAADF STEM observations (Figure 1d). The elemental compositions of the particles characterized by atom probe tomography (APT) showed that the particles were made of a core, enriched in Sc, and a thin Zr shell with a thickness of 1–2 nm (Figure 1e–f), which is consistent with their formation mechanism involving thermodynamic and kinetic processes [33]. Tomographic reconstruction from extra APT datasets revealed that the volume fraction of the particles was (7.80 ± 3.83) × 10^21^ m^−3^ in number density. The number density will be used for the following quantitative calculation.

### 3.2. Strain-Stress Curves and Microstructure Characteristics

Figure 2 indicates that the engineering strain-stress curves of the aged alloy without and with Al_3_Sc_1−x_Zr_x_ particles failed in air and 3.5 wt.% NaCl solution during SSRT; it also presents their corresponding microstructures. It can be seen that the Al_3_Sc_1−x_Zr_x_ particles improved the yield strength by 161 ± 7 MPa (40 ± 3 percent), and increased the ultimate tensile strength by 122 ± 4 MPa (27 ± 1 percent), respectively, while also enhancing ductility (increase from 13.5 ± 0.3% to 17.7 ± 0.1%) (Figure 2a). 

Equation (1) was used to calculate the I_scc_ (SCC sensitive indices).
(1)$$I_{\mathrm{SCC}}=1-\frac{P_{\mathrm{in}\;\mathrm{air}}}{P_{\mathrm{in}\;3.5\%\mathrm{NaCl}}}$$
where P_in air_ and P_in 3.5% NaCl_ are the measured properties (ultimate tensile strength (UTS), yield strength (YS), elongation to failure (El_f_) and fracture energy (FE)) in air and 3.5 wt.% NaCl solution, respectively. I_scc_, shown in Figure 2b, increased with the increase of SCC susceptibility. The SCC sensitive values were different when different properties were used. For example, the I_scc_ of UTS, YS, El_f_ and FE were 2.8 ± 0.2%, 2.6 ± 0.2%, 13.8 ± 0.7% and 21.4 ± 0.1% in the aged Al–Zn–Mg alloy, respectively, and 1.6 ± 0.1%, 0.3 ± 0.5%, 8.4 ± 0.1% and 12.7 ± 0.5% in the aged alloy containing particles, respectively. UTS and YS had a lower SCC susceptibility, while E_f_ and FE seemed to be more sensitive to SCC. Al_3_Sc_1−x_Zr_x_ particles can effectively reduce the I_scc_ of E_f_ and FE, indicating their positive effect on SCC resistance. 

Figure 2c,d show the corresponding fracture surfaces and the grain and grain boundary characteristics of the studied alloys without and with particles, respectively. For the alloys without particles (Figure 2c), obvious intergranular cracks, a high proportion of high angle grain boundaries, coarse grain boundary aging phases and widened precipitate free zone (PFZ) can be observed. This indicates that SCC tends to occur along high angle grain boundaries, due to their severe microchemistry inhomogeneity. In contrast to the alloy without particles, the fracture surface of the alloys with particles that failed in a corrosive environment was characterized by ductile transgranular cracking, and there were lots of small equiaxial dimples (Figure 2d). Moreover, an uncrystallized deformed structure was observed. The TEM images indicate that the coarse aging phase or PFZ is almost absent, or formed on the subgrain boundaries, showing the lower microchemistry inhomogeneity. According to our quantitative EBSD data, the Al_3_Sc_1−x_Zr_x_ particles were shown to be able to decrease the fraction of high angle grain boundaries from 87% to 31%, and lower the average misorientation angles from 34.4° to 16.9° in aged alloys. Introducing Al_3_Sc_1−x_Zr_x_ induced obvious grain refinement, and the gain size decreased from 49.5 μm to 11.7 μm. 

## 4. Discussion

### 4.1. Explanation of the Excellent Mechanical Properties of Al_3_Sc_1−x_Zr_x_ Nanoparticles

To determine the main strengthening mechanism of Al_3_Sc_1−x_Zr_x_ nanoparticles, the selected area diffraction (SAD) patterns of the aged alloys and the interactions between the Al_3_Sc_1−x_Zr_x_ particles and the defects (dislocations and grain/subgrain boundaries) were investigated; the results are shown in Figure 3. Characteristic superlattice spots from Al_3_Sc_1−x_Zr_x_ and metastable η’(MgZn_2_) phase were detected in the selected area diffraction (SAD) patterns, taken along the [001]_Al_ zone axes (Figure 3a). In addition, strong interactions between the Al_3_Sc_1−x_Zr_x_ nanoparticles and defects, such as dislocations (Figure 3b) and subgrain/grain boundaries (Figure 3c), were observed, indicating that these nanoparticles can prevent the occurrence of recovery or recrystallization during thermal-mechanical processes (hot extrusion and solution treatment). The statistical analysis results, obtained from superlattice dark field TEM images, indicated that Al_3_Sc_1−x_Zr_x_ nanoparticles had an average diameter of 10.3 ± 3.6 nm. 

Combined with the results from Section 2, it can be concluded that due to the core-shell structure, the Al_3_Sc_1−x_Zr_x_ nanoparticles have highly thermal stability, and thus, still retain a coherent relationship with the matrix. The low lattice misfits of the Al_3_Sc_1−x_Zr_x_ nanoparticles (1.14 ± 0.03 percent) cause them to have strong interaction with defects. On the one hand, the strong pinning force from Al_3_Sc_1−x_Zr_x_ nanoparticles on grain/subgrain boundaries can inhibit the migration of grain/subgrain boundaries during hot extrusion and heat treatment, leading to limited grain growth of aged Al–Zn–Mg alloys. Therefore, it was shown that Al_3_Sc_1−x_Zr_x_ nanoparticles decrease the gain size of aged Al–Zn–Mg alloys from 49.5 μm to 11.7 μm, generating Hall-Petch or grain refinement strengthening. On the other hand, Al_3_Sc_1−x_Zr_x_ nanoparticles can prevent dislocation motion (Figure 3b), producing precipitation strengthening. Due to the larger particle size of those particles (10.3 ± 3.6 nm), the precipitation strengthening occurs via the Orowan bypass mechanism, rather than shearing mechanism. Combined with the 3DAP, EBSD and TEM data, quantitative analyses were performed to verify which strengthening mechanism of Al_3_Sc_1−x_Zr_x_ particles is the dominant one, i.e., Orowan strengthening or gain/subgrain boundary strengthening.

#### 4.1.1. Orowan Strengthening

The increased yield strength from Orowan strengthening (Δσ_Or_) can be calculated by the following equations [29]:(2)$$\Delta \sigma_{\mathrm{Or}}=K_{4}M{\left(1-v\right)}^{-0.5}(\mathrm{Gb}/\lambda )\mathrm{In}(d_{s}/b)$$
(3)$$d_{s}=\frac{\pi d_{m}}{4}$$
(4)$$\lambda =\left[\frac{1}{2}{\left(\frac{2\pi }{3f_{V}}\right)}^{0.5}-1\right]\frac{\pi d_{m}}{4}$$
where K_4_ = 0.127, M (Taylor factor) = 3.06, v (Poisson’s ratio) = 0.331 and G (shear modulus) = 27.8 GPa, and b (Burgers vector) = 0.286 nm, d_s_ and λ are the mean particle diameter and an effective interparticle distance, respectively, and were obtained from the measured results. The calculated Orowan strengthening caused by the Al_3_(Sc_1−x_Zr_x_) nanoparticles is about 159 MPa. 

#### 4.1.2. Hall–Petch Strengthening 

As Al_3_(Sc_1−x_Zr_x_) nanoparticles can decrease the grain size, the generated Hall–Petch strengthening was obtained by Equation (5).
(5)$$\sigma_{H-P}=\sigma_{0}+kd^{-1/2}$$
where σ_0_ and k (0.04 MPa × m^1/2^ [34,35] in aluminum) reflect the intrinsic resistance of the aluminum lattice to dislocation motion and grain boundary migration. The increased yield strength from Hall–Petch strengthening (Δσ_H-P_) caused by Al_3_(Sc_1−x_Zr_x_) nanoparticles can be calculated by Equation (6):(6)$$\Delta \sigma_{H-P}=k(d_{\mathrm{with}\;\mathrm{particles}}^{-1/2}-d_{\mathrm{without}\;\mathrm{particles}}^{-1/2})$$

By substituting the k value and the measured grain sizes into the above equation, Δσ_H-P_ was calculated to be about 6 MPa. 

Therefore, the improved strength from Al_3_Sc_1−x_Zr_x_ particles is mainly due to their strong Orowan strengthening (159 MPa) and weak Hall-Petch strengthening (6 MPa). The calculated YS increment was about 165 MPa, close to the experimental value (161 MPa).

Except for high-strength, the studied alloy with particles showed admirable ductility. Figure 4 shows the property data for yield strength versus elongation to failure. The error bars in Figure 4 represent one standard deviation away from the mean. The data in this figure are compared with some typical commercially available Al alloys with high-strength (7xxx series Al alloys), medium-strength (6xxx series Al alloys), and high-ductility (2xxx series Al alloys) [1] based on our previous work [18] and parts of the laboratory data [16,20,36,37]. However, it is expected that some variability in mechanical properties exists in the published papers under the given laboratory conditions. It can be seen that the alloy with nanoparticles shows an excellent combination of strength and ductility. In addition, compared with the alloy without particles, significant improvements in strength and ductility were observed by introducing nanoparticles. The ductility in the alloy with particles may be ascribed to the lower microchemistry and microstructure inhomogeneity in the prevailing low angle grain boundaries. 

### 4.2. Interpretation of the Enhanced SCC Resistance by Al_3_Sc_1−x_Zr_x_ Nanoparticles

To further understand the improved resistance of SCC with Al_3_Sc_1−x_Zr_x_ particles, two aspects resulting in corrosion initiation and crack propagation in aluminum alloys should be considered. First, the second-phases or microchemistry heterogeneity lead to the concentration of electrochemical reactions, thereby accelerating SCC initiation. It has been reported that in aluminum alloys, phases or defects below ~4 nm in diameter do not generate localized corrosion [38]. Therefore, the metastable phase η’ (MgZn_2_) basically has no effect on SCC corrosion initiation. In addition, Al_3_Sc_1−x_Zr_x_ particles and equilibrium phase η (MgZn_2_) phases on the grain boundaries are potentially dangerous phases for SCC. Wloka [39] and Cavanaugh [40] pointed out that Al_3_Sc_1−x_Zr_x_ or Al_3_Sc particles have no negative effect on corrosion due to their slightly cathodic role on α-Al. However, the corrosion potentials of the grain boundary phase (MgZn_2_), PFZ and the matrix of the high strength aluminum alloys were close to −0.86, −0.57 and −0.68 V [41], respectively, revealing that the coexistence of coarse MgZn_2_ and widened PFZ can accelerate the galvanic reaction at the adjacent periphery of the grain boundaries. According to our quantitative EBSD data, Al_3_Sc_1−x_Zr_x_ particles can decrease the fraction of high angle grain boundaries from 87% to 31%, and lower the average misorientation angles from 34.4° to 16.9°. Figure 2 indicates that coarse MgZn_2_ and PFZ generally do not form along low-angle grain boundaries. As a consequence, it can be concluded that low angle grain boundaries can decrease the microchemistry difference or microgalvanic corrosion and inhibit stress corrosion cracking initiation. Secondly, as revealed from the potentiodynamic polarization curve and electrochemical impedance spectroscopy (Figure 5), the alloys with a low proportion of high angle grain boundaries reduce the anodic kinetics and dissolution kinetics of the Al–Zn–Mg alloy, decelerating the stress corrosion cracking propagation. Therefore, the low microchemistry heterogeneity of the prevalent low angle grain boundaries associated with the substructure is the critical factor for the reduced SCC susceptibility of the alloy with Al_3_Sc_1−x_Zr_x_ particles.

## 5. Conclusions

The nanostructure characteristics of Al_3_Sc_1−x_Zr_x_ nanoparticles and their effects on the mechanical properties and SCC behavior of Al–Zn–Mg alloys were investigated by 3D atom probe analyses, electron microscopy methods, electron back scattered diffraction techniques, electrochemical measurements and slow strain rate tests. Three main conclusions were drawn from the experimental investigations:

Adding a small amount of Sc (0.10 weight%) and Zr to Al–Zn–Mg alloys can form relatively high-density secondary L1_2_-Al_3_(Sc_1−x_Zr_x_) particles with a volume fraction of (7.80 ± 3.83) × 10^21^ m^−3^. After hot extrusion and solution-quenching-aging treatment, Al_3_(Sc_1−x_Zr_x_) particles can retain a fully coherent relationship with the matrix (a low lattice misfit of 1.14 ± 0.03%), a nanosize (10.3 ± 3.6 nm), and a stable core-shell L1_2_-nanostructure in Al–Zn–Mg alloys. 

In T6 Al–Zn–Mg alloys, Al_3_(Sc_1−x_Zr_x_) the nanoparticles themselves can generate strong Orowan strengthening (the calculated increased yield strength value of Al–Zn–Mg alloy was 159 MPa), and those particles have a certain pinning force on subgrain/grain boundaries to inhibit the grain growth, producing weak Hall-Petch strengthening (the calculated increased yield strength value is 6 MPa). The calculated yield strength increment value based on the nanostructure characteristics of Al_3_Sc_1−x_Zr_x_ nanoparticles was 165 MPa, close to the experimental value (161 ± 7 MPa). Meantime, the alloy with particles had better ductility (elongation is 17.7%) than the Sc-free Al–Zn–Mg alloys (elongation was 13.5%).

The failure of the Sc-free Al–Zn–Mg alloy in a corrosive environment was predominantly caused by intergranular cracking, coarse grain boundary precipitates and widened PFZ. Al_3_Sc_1−x_Zr_x_ nanoparticles can change the fracture mechanism of Al–Zn–Mg alloys in a 3.5% NaCl solution from intergranular cracks into transgranular failure, and decrease the proportion of high-angle grain boundaries, with higher microchemistry inhomogeneity than subgrain boundaries, from 87% to 31%, thereby reducing the microchemistry differences around the grain boundaries and anodic dissolution kinetics, and improving intergranular SCC resistance and ductility. 

The new Al–Zn–Mg alloy with Al_3_Sc_1−x_Zr_x_ nanoparticles showed an excellent combination of strength and ductility and low SCC susceptibility.

## Figures and Tables

**Figure 1 materials-13-01909-f001:**
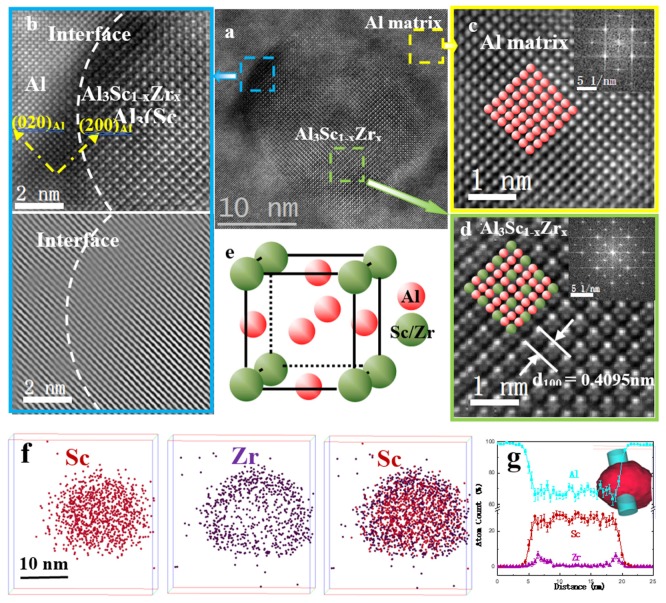
High-resolution HAADF STEM images and 3DAPT dataset confirming the Al_3_Sc_1−x_Zr_x_ nanoparticles with an ordered L1_2_ core-shell structures and full lattice coherence. (**a**) High-resolution STEM images of Al_3_Sc_1−x_Zr_x_; (**b**–**d**) the filtered HRTEM HAADF STEM images of the interface, α(Al) matrix, nanoparticles, respectively, along with the inset showing fast Fourier transform (FFT) patterns; (**e**) crystal structure of L1_2_-Al_3_Sc_1−x_Zr_x_; (**f**) Atom probe tomography reconstruction of the particles; (**g**) atom distribution across the spherical particles along the blue bar as the inset.

**Figure 2 materials-13-01909-f002:**
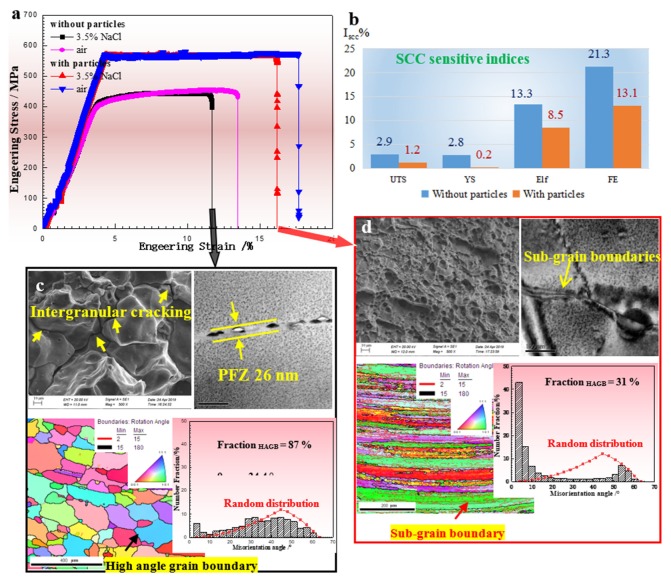
SSRT results of the aged alloy with and without particles and the corresponding microstructures. (**a**) Tensile curves; (**b**) SCC sensitive factors; (**c**,**d**) SEM fracture surface, EBSD and bright-field TEM images of the alloys failed in corrosive environment.

**Figure 3 materials-13-01909-f003:**
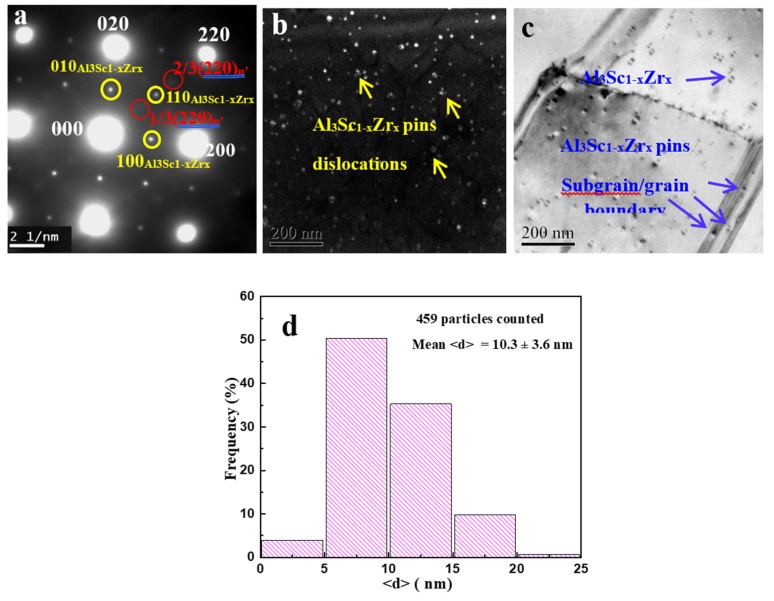
TEM results of precipitates in aged alloys. (**a**) SAD patterns, (**b**) (110) superlattice centered dark field image, showing the interaction between Al_3_Sc_1−x_Zr_x_ particles and dislocations, (**c**) bright-field TEM image, showing the interaction between Al_3_Sc_1−x_Zr_x_ particles and grain/subgrain boundaries, (**d**) size distribution histograms of Al_3_Sc_1−x_Zr_x_ nanoparticles.

**Figure 4 materials-13-01909-f004:**
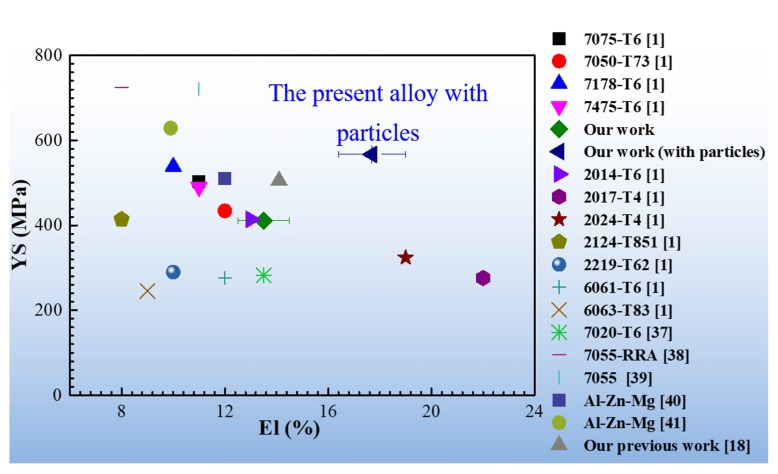
Yield strength versus elongation to failure.

**Figure 5 materials-13-01909-f005:**
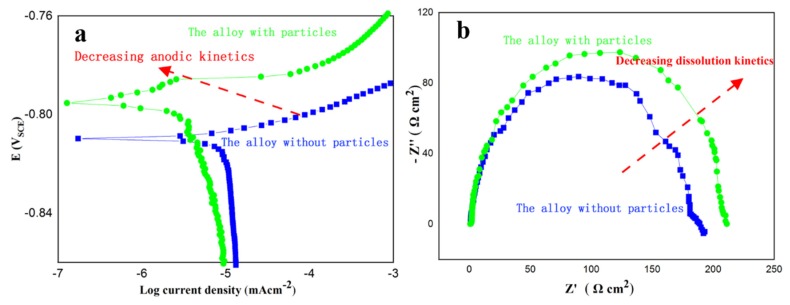
Electrochemical behavior of the alloys without and with particles. (**a**) Potentiodynamic polarization curve. (**b**) Nyquist plots from electrochemical impedance spectroscopy.

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
