# Peer review of "Nanostructure Characteristics of Al3Sc1−xZrx Nanoparticles and Their Effects on Mechanical Property and SCC Behavior of Al–Zn–Mg Alloys"

_materials, 2020, doi:10.3390/ma13081909_

Round 1

Reviewer 1 Report

All comments are in a positive review.

Nano-structure characteristics of Al3Sc1−xZrx nanoparticles and their effects on mechanical property and  SCC behaviour of Al-Zn-Mg alloys

Ying Deng, Ziang Yang  and Guo Zhang   

The article continues the study  properties of the  Al-Zn-Mg alloys and characteristics of Al3Sc1−xZrx nano-particles and their effects on mechanical property and SCC behavior of Al-Zn-Mg alloys/.

  In this article, the authors present new   idea for the simultaneous improvement in mechanical property and corrosion resistance of high-strength Al-Zn-Mg  alloys.

The article contains new original results and matches the profile of the journal

The article is written in clear, understandable language.

Useful publication on a relevant topic

Author Response

Response to Reviewer #1:

We really appreciate your kind comments. It gives a good incentive for us to continue our research work.

Reviewer 2 Report

The subject of this work is interesting however the manuscript can not be accepted in its present form due to the following major points: (The whole manuscript has to be revised and rewritten carefully concerning the following comments)

  1. The main objective and starting point of this research work is not clear and lacks novelty. I mean, from where you start your work according to other previous works.
  2. The introduction has to be rewritten carefully including more details relating to your materials (why selected) and the engineering application that required to apply SCC study. (Including the first comment)
  3. The experimental work section is not clear enough to the readers. The authors have to add more details concerning the fabrication of alloys, the instrument used for SCC test, polarization curves, electron microscopy used for microstructure characterization, etc. This section is too important to be more clear for the readers of Materials Journal.
  4. More explanations (with references) of results should be scientifically in-deep and in logic order that clear the main aims of this work. Speculative discussions are not helpful.
  5. Conclusions should be in form of specific points including the most significant and new conclusions drawn from this work.

Author Response

Comments and Suggestions for Authors from Reviewer # 2: 

The subject of this work is interesting however the manuscript can not be accepted in its present form due to the following major points: (The whole manuscript has to be revised and rewritten carefully concerning the following comments)

Response to Reviewer #2:

Thank you very much for your comments, and it is really helpful for us to improve our manuscript. We have revised our paper according to your comments and suggestion, and the detailed responses are shown as follows:

Question 1.

The main objective and starting point of this research work is not clear and lacks novelty. I mean, from where you start your work according to other previous works.The introduction has to be rewritten carefully including more details relating to your materials (why selected) and the engineering application that required to apply SCC study. (Including the first comment)

Answer:

  • Thanks very much for your suggestion, and we have rewritten the Introduction section. In the revised introduction, the research background was listed in detail, relating to your materials (why selected) and the engineering application that required to apply SCC study, and the main objective and starting point of this research workwere further highlighted.

Question 2. 

The experimental work section is not clear enough to the readers. The authors have to add more details concerning the fabrication of alloys, the instrument used for SCC test, polarization curves, electron microscopy used for microstructure characterization, etc. This section is too important to be more clear for the readers of Materials Journal. 

Answer:

  • Thanks very much for your suggestion. We supplemented more details about the  fabricationprocess of alloys and added the instrument used for SCC test. The instrument types or models, applied for SEM/EBSD (ZEISS EVO10-3412 field emission gun scanning electron microscope , operating at 20 kV), HRTEM/HAADF-STEM (a Titan G2 60–300 high-resolution aberration- corrected transmission electron microscope (HRTEM), operating at 300 kV) , and 3DAP (A LEAP 4000X HR under UV laser pulsing was used for three-dimensional atom probe tomography (3DAPT)), have been provided in the previous uploaded version.

Question 3.  

More explanations (with references) of results should be scientifically in-deep and in logic order that clear the main aims of this work. Speculative discussions are not helpful.

Answer:

  • Our discussions were divided into two parts. Part 1 is about the strengthening mechanism. The increment in yield strength of Al-Zn-Mg alloys caused by Al3Sc1−xZrxnano-particles was calculated quantitatively based on our 3DAP, EBSD and TEM results. Part 2 is about the improved SCC resistance from Al3Sc1−xZrx nano-particles. This discussion is based on the comparative microstructural results between the alloy without and with particles. May be our language expresion is not too detailed to give you a speculative We added some text to further illustrate and disscuss our results. Thank you for your suggestion.   .

Question 4.  

Conclusions should be in form of specific points including the most significant and new conclusions drawn from this work.

Answer:

  • Thank you very much for your comments. We have revised our conclusions into the form of specific points including the most significant and new conclusions drawn fromour work.

Reviewer 3 Report

While the stated properties of yield strength and elongation for a 7xxx type aluminum alloy with low amounts of Sc added to form L12 precipitates are an improvement over the presented references, there is an extreme lack of context regarding the established literature for such precipitates. The current work is presented against commercial alloys, and no other experimental alloys of the same type (utilizing Al3ZrSc) – only works containing “high” amounts of Sc are cited in the introductions, when there are numerous other works are similar levels of Sc to this paper (~0.1Sc). Further, there is nothing novel in the characterization of these precipitates that provides adequate context for why the author’s work is an improvement over other work in this field, including their own previous works.

The statement that “the nano-structure of Al3ScZr is still lack of detailed investigation” is untrue, there are atom probe and TEM studies describing the core-shell structure of Al3ScZr L12 precipitates as early as 2004:

  • https://doi.org/10.1016/j.scriptamat.2004.03.033
  • https://doi.org/10.1016/j.scriptamat.2004.11.021

and more recently even modeling to confirm the energetic balance of the core-shell and other segregation: https://doi.org/10.1016/j.matchar.2019.109898

Regarding “Simultaneous improvement in mechanical property and corrosion resistance” as depicted in Fig 4, there are numerous mechanical property studies on Al-Zn-Mg-Sc alloys that are not cited and, while the present work may have higher elongation/YS, it should be benchmarked against these works. In particular, there must be a discussion of how this characterized microstructure is different than previous works that lead to these improved mechanical properties

  • For example, in this work, 721MPa/11% would fall outside of the draw region on Fig 4 (but is still a lower ductility) https://doi.org/10.1016/j.jallcom.2015.10.296
  • This work also falls outside of the region depicted in Fig 4 for some of the aging conditions and is low Sc https://doi.org/10.1016/j.jallcom.2007.02.054
  • This work has lower properties but describes the core/shell structure and is still low Sc https://doi.org/10.1016/j.msea.2014.08.041
  • The authors do not even include their earlier works on this subject on the plot, eg. Ref [21] and describe the improvements they have made

These are just a few of the low Sc studies which are not referenced in this paper.

There is not actually a SCC study in this, only a brief discussion of sensitivity factors – so the title and abstract are somewhat misleading. Further, there aren’t any error bars on Fig 2B, so it is not clear that the “with particles” factors are actually and significant improvement over the “without particles”. Similarly, the relative improvement of the “with particles” in the various conditions (yield, uts, elongation) is given no context in the relationship to SCC resistance. Perhaps this is in the testing specification cited, but it is not easy to access, and should be discussed in the paper. For example, does the fact that the elongation property is more sensitive to SCC mean something regarding the severity or mode of SCC versus the UTS being sensitive?

Author Response

Comments and Suggestions for Authors from Reviewer # 3:

Question 1.  

While the stated properties of yield strength and elongation for a 7xxx type aluminum alloy with low amounts of Sc added to form L12 precipitates are an improvement over the presented references, there is an extreme lack of context regarding the established literature for such precipitates. The current work is presented against commercial alloys, and no other experimental alloys of the same type (utilizing Al3ZrSc) – only works containing “high” amounts of Sc are cited in the introductions, when there are numerous other works are similar levels of Sc to this paper (~0.1Sc). Further, there is nothing novel in the characterization of these precipitates that provides adequate context for why the author’s work is an improvement over other work in this field, including their own previous works.

Answer:

  • Thank you very much for your comments. We have rewritten the Introduction section, including the illustration of work context and the novelty of the characterization of particles.

Question 2.  

The statement that “the nano-structure of Al3ScZr is still lack of detailed investigation” is untrue, there are atom probe and TEM studies describing the core-shell structure of Al3ScZr L12 precipitates as early as 2004:

https://doi.org/10.1016/j.scriptamat.2004.03.033

https://doi.org/10.1016/j.scriptamat.2004.11.021

and more recently even modeling to confirm the energetic balance of the core-shell and other segregation: https://doi.org/10.1016/j.matchar.2019.109898

Answer:

  • Thank you very much for your comments. We have added the above three papers into our revised paper. And what’s worth mentioning here is that unlike the previous reports about the characterization of Al3Sc1-xZrxdispersoids, which either focused on composition distribution alone, or on their structure, or morphology separately, in this paper all characteristics of Al3Sc1-xZrx nano-particles including atomic distribution, lattice arrangement, the degree of coherence with matrix, number density and size, were all taken into consideration. Furthermore, the relationship between the above nano-structure characteristics of those particles and mechanical properties/SCC behaviour was established. We have revised our abstract and introduction section to avoid to the misleading understanding.

Question 3.  

Regarding “Simultaneous improvement in mechanical property and corrosion resistance” as depicted in Fig 4, there are numerous mechanical property studies on Al-Zn-Mg-Sc alloys that are not cited and, while the present work may have higher elongation/YS, it should be benchmarked against these works. In particular, there must be a discussion of how this characterized microstructure is different than previous works that lead to these improved mechanical properties

For example, in this work, 721MPa/11% would fall outside of the draw region on Fig 4 (but is still a lower ductility) https://doi.org/10.1016/j.jallcom.2015.10.296

This work also falls outside of the region depicted in Fig 4 for some of the aging conditions and is low Sc https://doi.org/10.1016/j.jallcom.2007.02.054

This work has lower properties but describes the core/shell structure and is still low Sc https://doi.org/10.1016/j.msea.2014.08.041

The authors do not even include their earlier works on this subject on the plot, eg. Ref [21] and describe the improvements they have made

These are just a few of the low Sc studies which are not referenced in this paper.

Answer:

  • Thank you very much for your comments. Fig. 4 is to illustrate that the Al-Zn-Mg alloy with particles have a good combination of strength and ductility in this study. The comparision data are from the alloys produced by the similar method, and in the similar as-heat-treated condition. Compared with the mechanical properties of our alloys (YS: 567MPa, El:17.7%), the alloys in the paper “Microstructure evolution and mechanical properties of an ultrahigh strength Al-Zn-Mg-Cu-Zr-Sc (7055) alloy processed by modified powder hot extrusion with post aging (https://doi.org/10.1016/j.jallcom.2015.10.296) ”have a lower elongation (11%), and the mechanical properties is close to the 7055-RRA (listed in Fig.4 of our manuscipt). Moreover, the alloys in (https://doi.org/10.1016/j.jallcom.2015.10.296)is produced by powder metallurgy, which is different from our studied alloys and most of commercial aluminum alloys.

       The mechanical properties of the alloys under different homogenization treatment in the paper titled “Effects of homogenization treatment on recrystallization behavior and dispersoid distribution in an Al–Zn–Mg–Sc–Zr alloy(https://doi.org/10.1016/j.jallcom.2007.02.054)”are present in the following Figure. It can be found that the yield strength and elongation to failure of the Al–Zn–Mg–Sc–Zr alloys under different conditions are all inferior to our studied alloys with particles.      

     (This Figure was uploaded in the PDF.)

The mechanical properties of the alloys reported in the paper “https://doi.org/10.1016/j.msea.2014.08.041” (Effect of Sc/Zr ratio on the microstructure and mechanical properties of new type of Al–Zn–Mg–Sc–Zr alloys) were shown in the following table. It can be found that the elongation of the aged alloys (7.9-8.6%) is much lower than the alloys we studied.  

    (This table was uploaded in the PDF.)

Moreover, the nano-structure characteristics of Al3Sc1-xZrx nano-particles includes core/shell structure, but is not limited to core/shell structure. Lattice arrangement, the degree of coherence with matrix, number density and size of Al3Sc1-xZrx nano-particles were investigated in our study and they have different effects on mechanical properties and SCC behaviour of Al-Zn-Mg alloys, which can not be ignored.

In our paper, it clearly indicates that “The data in figure 4 are compared with some typical commercially available Al alloys with high-strength (7xxx series Al alloys), medium-strength (6xxx series Al alloys), and high-ductility (2xxx series Al alloys) [29-31]”. The mechanical properties in Fig.4 just list the comparison between the studied alloy with particles with commercially available Al alloys, but there is expected to be some variability in the actual values in the published literature depending on the testing conditions of a given laboratory.

Finally, combined with your suggestion and comments, we decided to add the above references and our previous work, mentioned by reviewers, into Fig.4. Moreover, the corresponding instructions have been added and the space line in Fig. 4 was deleted. Thank you very much for your patient and careful guidance.

Question 4.  

There is not actually a SCC study in this, only a brief discussion of sensitivity factors -so the title and abstract are somewhat misleading. Further, there aren’t any error bars on Fig 2B, so it is not clear that the “with particles” factors are actually and significant improvement over the “without particles”. Similarly, the relative improvement of the “with particles” in the various conditions (yield, uts, elongation) is given no context in the relationship to SCC resistance. Perhaps this is in the testing specification cited, but it is not easy to access, and should be discussed in the paper. For example, does the fact that the elongation property is more sensitive to SCC mean something regarding the severity or mode of SCC versus the UTS being sensitive?

Answer:

  • SCC performance of the studied alloys were evaluated by slow strain rate tests (SSRT) and the sensitivity factors reflect the degree of SCC according to GB 15970.7-2000 [31, 32] and ASTM G129-00 [33] in our paper. The ref [31] is GB 15970.7-2000 “National Standard of China, Corrosion of Metals and Alloys- Stress Corrosion Testing – Slow Strain Rate Testing”, the ref [32] is a typical paper about SCC, “ Characterization of the SCC behaviour of 8090 Al–Li alloy by means of the slow-strain-rate technique’, and the ref [33] is “Standard practice for slow strain rate testing to evaluate the susceptibility of metallic materials to environmentally assisted cracking.” Moreover, the same method (slow strain rate tests) used for evaluate SCC resistance of aluminum alloys can be found in many literature, such as the following reference [1-3]. Therefore, the SSRT method and the sensitivity factors used for SCC evaluation of our studied aluminum alloys in this paper are valid.
  • G. Song, W. Dietzel, B.J. Zhang, W.J. Liu, M.K. Tseng, A. Atrens, Stress corrosion cracking and hydrogen embrittlement of an Al-Zn-Mg-Cu alloy, Acta Mater. 52 (2004) 4727-4743. https://doi.org/10.1016/j.actamat.2004.06.023.
  • L. Ou, J.G. Yang, M.Y. Wei, Effect of homogenization and aging treatment on mechanical properties and stress-corrosion cracking of 7050 alloys, Metall. Mater. Trans. A Phys. Metall. Mater. Sci. 38 (2007) 1760–1773. https://doi.org/10.1007/s11661-007-9200-z.
  • Deng, Z. Yin, K. Zhao, J. Duan, J. Hu, Z. He, Effects of Sc and Zr microalloying additions and aging time at 120°C on the corrosion behaviour of an Al-Zn-Mg alloy, Corros. Sci. 65 (2012) 288–298. https://doi.org/10.1016/j.corsci.2012.08.024.

      About the error bars in Fig 2b, SCC sensitivity factors in Fig. 2b are the calculated values according to Eq. (1), not the measured experimental values. Therefore, the error bars are not present.  

      Due to the fact that it is not easy to access to the UTS, YS and El data in figure 2b, we added some relevant text description about SCC data into the revised paper.

      About the question “For example, does the fact that the elongation property is more sensitive to SCC mean something regarding the severity or mode of SCC versus the UTS being sensitive?”, there is no relevant research and maybe it can be a new research direction for our future work. In international standards, the elongation is generally used to calculate the sensitivity factor for stress corrosion. The main reason is that the calculated sensitivity factor of elongation is usually higher than the calculated sensitivity factor of strength.

Round 2

Reviewer 2 Report

The authors did all required modifications. The manuscript can be accepted in its present form.

Author Response

Comments and Suggestions for Authors from Reviewer # 2: 

The authors did all required modifications. The manuscript can be accepted in its present form.

Response to Reviewer #2:

We really appreciate your kind comments. Thank you again for your patient and careful guidance.

Reviewer 3 Report

Thank you for rewriting the introduction and the literature comparison section of the discussion, you have greatly improved the context of your work in terms of the broader literature.

Regarding original Question 4, the SCC sensitivity factors, the reviewer is not questioning the validity of the SSRT tests for use in assessing SCC sensitivity as the authors response indicates, only that there is no contextual discussion of the sensitivity factors. Being unfamiliar with the particular specification used, the question was whether there may be some relation where the %property change correlates with SCC behavior, for example if the %property change is greater than a certain percent, does that indicate the material is then considered “susceptible” to SCC whereas below a certain percent it may not be susceptible?

  • Given that equation (1) uses measured values that should have error bars, the assertation that the RA…the Acta paper cited Ref [1] in the author’s response has error bars on all the RA calculations
  • Response states that elongation is usually used as the factor, OK then is your elongation decrease better or worse than other aluminum efforts? Again looking at Ref [1], their elongation loss is similar or lower (in the second peak-aged state)

It’s still an open question to be that this alloy/microstructure will be better for SCC than others.

Author Response

Please see the attachment。
